# Deep Spatiotemporal Model for COVID-19 Forecasting

**DOI:** 10.3390/s22093519

**Published:** 2022-05-05

**Authors:** Mario Muñoz-Organero, Paula Queipo-Álvarez

**Affiliations:** Telematic Engineering Department, Universidad Carlos III de Madrid, 28911 Madrid, Spain; pqueipo@it.uc3m.es

**Keywords:** machine learning, deep learning, COVID-19 forecasting, spatiotemporal model, model optimization

## Abstract

COVID-19 has caused millions of infections and deaths over the last 2 years. Machine learning models have been proposed as an alternative to conventional epidemiologic models in an effort to optimize short- and medium-term forecasts that will help health authorities to optimize the use of policies and resources to tackle the spread of the SARS-CoV-2 virus. Although previous machine learning models based on time pattern analysis for COVID-19 sensed data have shown promising results, the spread of the virus has both spatial and temporal components. This manuscript proposes a new deep learning model that combines a time pattern extraction based on the use of a Long-Short Term Memory (LSTM) Recurrent Neural Network (RNN) over a preceding spatial analysis based on a Convolutional Neural Network (CNN) applied to a sequence of COVID-19 incidence images. The model has been validated with data from the 286 health primary care centers in the Comunidad de Madrid (Madrid region, Spain). The results show improved scores in terms of both root mean square error (RMSE) and explained variance (EV) when compared with previous models that have mainly focused on the temporal patterns and dependencies.

## 1. Introduction

The spread of COVID-19 has attracted significant attention by the research community since its early cases appeared in late 2019 in Wuhan, China [1]. The rapid human-to-human transmission of the disease has generated waves of infections around the globe that have strained heath care systems to their limits [2]. For this reason, the development of predictive models capable of forecasting the dynamics of the virus on a short and long-term range is playing a crucial role to mitigate the effects of the spreading of the virus, implement optimal policies, and optimize the use of health care resources [3].

Different types of predictive models have been used in order to forecast COVID-19 infections, recoveries and deaths such as epidemic, mathematical, statistical, machine learning-based and hybrid models. The most applied epidemic models are the Susceptible, Infected, and Recovered (SIR) model and the Susceptible, Exposed, Infected, and Recovered (SEIR) model [4]. Both SIR and SEIR models group people in different compartments and solve differential equations to move people among compartments, modeling the spread of the disease. These differential equations describe the variation over time of the amount of people that get infected after being exposed to the virus and finally either recover or die from it based on the amount of people susceptible and already infected and the dynamics of the virus captured as rate parameters. Statistical models describe the spread of the disease in terms of stochastic variables which can be modeled using probability functions, some of which can be observed while others can be estimated. The authors in [5] describe the concept of statistical models for COVID-19 and use data to fit probability distributions to stochastic variables defining the spread of the COVID-19 virus such as the time to develop symptoms and the time to require hospitalization. Machine learning models try to learn patterns in the observed data by training models that can learn from data. Once trained, the models could be used to predict the outcome for new data not currently seen by the model. The authors in [6] published a survey on several machine learning models that have been used for COVID-19 predictions including deep learning models. Compared with epidemic models, machine learning models do not have to generate a simplified system in order to characterize the spreading of the virus but to observe data samples coming from different sensors and learn from them. However, machine learning models require previous data in order to fit the weights inside them. This is even more significant for deep machine learning models, which are able to learn more intricate patterns from data samples but have to find optimal values for a higher number of weights, which require having enough data to train them. Epidemic and machine learning models have also been combined into hybrid models which use machine learning models in order to find values from data for the different parameters in the epidemic models. The authors in [7] implemented and evaluated a framework with machine learning models trained to extract epidemic dynamics from the infection data to improve a county-level spatiotemporal epidemiological model that combines a spatial Cellular Automaton (CA) with a temporal susceptible-undiagnosed-infected-removed (SUIR) model. Although hybrid models are able to tune and optimize results for epidemic models, they will require the availability of data and will be limited to the simplifications made by the epidemic model.

This paper proposes a novel deep learning model for COVID-19 forecasting that combines spatiotemporal information in order to provide estimations about how the virus will migrate inside an area over time. The model generates sequences of infection maps from georeferenced COVID-19 reported data and makes use of both Convolutional Neural Networks (CNN) and Recurrent Neural Networks (RNN) to estimate the spread of the virus for each region in the map. The model is validated using the sensed and reported data from the Madrid region in Spain [8]. The major contribution of the proposed model is that it is able to use the combined space and time information together over a region outperforming previous models found in the literature when using the same dataset.

The manuscript is organized as follows. Section 1, this section, introduces the foundations and objectives of the research in this paper. Section 2 reviews the major results in previous studies to focus the new contributions of the current research. Section 3 captures the description and details of the model proposed as well as the dataset used to validate it. Section 4 captures the parameter optimization process for the model and the validation of results, including a comparison with previous published models based on deep learning approaches for COVID-19 forecasting. Finally, Section 5 captures some conclusions of the research in this paper.

## 2. Related Work

Machine learning models have been used in different areas related with the COVID-19 pandemic such as to help clinicians in the diagnoses and prognoses of the COVID-19 disease, to estimate the risk of getting infected and to help local authorities to assess optimal policies based on the forecasting of the spread of COVID-19 cases.

The authors in [9] proposed learning algorithms which include logistic regression, decision tree, support vector machines, naive Bayes, and artificial neutral network using an epidemiology labeled dataset for predicting the positive and negative COVID-19 cases of Mexico. Assaf et al. in [10] also used machine learning models in order to estimate the risk for developing critical COVID-19. A comparison of deep learning approaches to predict COVID-19 infection is presented in [11]. The authors in [12] presented an artificial-intelligence technique based on a deep convolutional neural network (CNN) to detect COVID-19 patients using real-world datasets examining chest X-ray images to identify such patients.

The majority of previous machine learning models able to forecast the evolution of the incidence of COVID-19 cases are based on the analyses of incidence-based time series. The authors in [13] present a comparative analysis of machine learning and soft computing models to predict the COVID-19 outbreak as an alternative to susceptible–infected–recovered (SIR) and susceptible-exposed-infectious-removed (SEIR) models. The authors use multi-layered perceptron (MLP) and an adaptive network-based fuzzy inference system (ANFIS) based on data from previously confirmed cases. Majhi et al. in [14] use regression-based, decision tree-based, and random forest-based models that have been built on the data from China and validated on India’s sample for predicting the evolution of COVID-19 using incidence data. The authors in [15] proposed the use of deep learning methods to learn from the previous data in the confirmed cases time series for COVID-19 and showed that through using Recurrent Neural Networks (RNN) based on bi-directional Long-Short Term Memory (Bi-LSTM), they were able to optimize previous results based on Support Vector Regression (SVR) and AutoRegressive Integrated Moving Average (ARIMA) models.

The spread of the virus not only has a temporal component but the movements of contagious people on a certain spatial region highly contribute to the evolution of the pandemic. The authors in [16] analyzed data from three different countries in a combined model which applied a Convolutional Neural Network (CNN) to help capture the influence of the movements of people among those countries in the estimation of future infection cases. Other studies have used Graph Neural Networks (GNN) in order to model the spatial spread of the virus into a machine learning model. The adjacencies among spatial locations are modeled either at the first layer of the model [17] or are used after a Recurrent Neural Network-(based) model has extracted the time component in order to exchange information among different zones [18].

Several recent studies have performed a comparison among the different machine learning methods used to forecast future cases on COVID-19 based on historic data. The authors in [19] performed a comparative study of machine learning methods for COVID-19 transmission forecasting, analyzing the data in several countries and doing a short-term prediction (one day). The paper compared shallow techniques based on Logistic Regression (LR) and Support Vector Machines (SVM) with deeper architectures such as Long-Short Term Memory (LSTM), Restricted Boltzmann Machines (RBM) and Convolutional Neural Networks (CNN). The study used the cumulative cases information and computed several metrics such as the root mean squared error (RMSE), the mean absolute error (MAE), the R-Squared (R2), the explained variance (EV) and the mean absolute percentage error (MAPE), showing that deep learning models outperformed the accuracy of shallow methods when used for short-term predictions. Alassafi et al. [20] performed a similar comparison over a one-week prediction horizon (PH) studying different configurations for a Recurrent Neural Network (RNN) showing that the network should be trained for at least 100 epochs.

This paper proposes and validates a new model based on the analyses of the time evolution of 2D COVID-19 infection maps that expand previous studies. The sensed data on new cases per week reported by each health care center in a region are geolocated and added into a 2D infection image. A CNN is applied to extract spatial patterns on the infection images. The result of the CNN in then fed into an RNN that extracts the time patterns of the infection images in order to estimate the evolution of the spread of the virus for each point in the map. The new model is validated with the data for all the heath areas inside the Madrid region for the sensed data since the second wave of COVID-19, showing promising results.

The spread of the COVID-19 virus has affected different behavioral aspects of society. Lockdowns and social policies have affected areas such as mobility in order to minimize social interactions. The authors in [21] analyzed the changes in micro-mobility usage before and during the lockdown period exploiting high-resolution micro-mobility trip data collected in Zurich, Switzerland. Specifically, docked bike, docked e-bike, and dockless e-bike were evaluated and compared from the perspective of space, time and semantics. The authors showed that the spread of the COVID-19 virus affected mobility policies, and at the same time, modified mobility patterns affected the spread of the virus. There are some studies such as [22,23] that use machine learning models in order to estimate traffic patterns which in turn could be used to enhance COVID-19 predictions. The study in [22] focuses on the construction of an effective solution designed for spatiotemporal data to predict the traffic state of large-scale traffic systems, while the authors in [23] proposed a framework based on the multi-layer perception (MLP) and Long Short-Term Memory (LSTM) model which integrate traffic incident-related factors and real-time traffic flow parameters to predict the residual traffic incident duration. Adding mobility data to the model proposed in this manuscript could potentially improve its results and will be described in future publications.

As previously captured in the introduction section, a different method from the one proposed in this manuscript for COVID-19 forecasting is based on the use of epidemic models. These models have been implemented in simulators such as [24], which introduced a simulation-based model to forecast the spreading behavior of the COVID-19 based on Saudi real data. The model we propose in this manuscript will also be enhanced to help with the estimation of the underlying parameters required to tune COVID-19 simulators as a future publication.

## 3. Materials and Methods

### 3.1. COVID-19 Forecasting Model

The spread of the COVID-19 virus is a dynamic process that is affected by the person-to-person exposure to infected cases [4] and therefore integrates the movement patterns of infected people over time with the temporal evolution of the symptom development and incubation process [17,18]. Many countries have imposed confinements to people in order to minimize the spatial movements and therefore mitigate the spread of the virus. However, some restrictions did not apply to workers that had to commute between close by regions on a daily basis and therefore contributing to the spread of the virus over regions.

Previous machine learning models applied to estimate the evolution of COVID-19 cases over the next days and weeks have mainly focused on the temporal patterns of the data [9,10,11,12,13,14,15,16] even in the case of applying deep learning models. Only a few models have included the spatiotemporal combined influence [17,18] but using a low granularity for the spatial data. More detailed COVID-19 infection maps have been successfully used in [7] as a basis to estimate some parameters to control epidemiological models. This section captures a new model that combines a complementary exploration of spatial COVID-19 infection maps with the extraction of time patterns. The model is captured in Figure 1.

The model in Figure 1 receives the incidence images for a particular region where people commute on a daily basis for the last *n* weeks and forecasts the expected new cases in a prediction horizon (PH) over the area. Figure 2 captures as an example an incidence image for the Madrid region in Spain during the first week of January 2022 using the data in [8]. The data in [8] provide information about the new cases reported weekly by each primary care health center. There are 286 primary care health centers in the Madrid region, and each of them covers a particular area in the region. Figure 2 shows the mapping of the 286 health areas into a grid image. The number of pixels in the image (adding the cases measured by each health center inside each geographical pixel) is configurable and optimizable in order to adapt to the movement patterns in each part of the world. The model in Figure 1 applies a CNN model to extract the spatial component affecting the spread of the virus over the region under study. The number of filters in the CNN and the number of output neurons are parameters that can be tuned to optimize results according to the particular characteristics of the movement patterns of people in that region. Therefore, each incidence image will be summarized into a number of output neurons that will be fed into a Recurrent Neural Network (RNN) based on Long-Short Term Memory (LSTM) cells. The RNN will learn the time patterns of the data capturing the spread of the virus over time (incubation and transmission processes). The number of memory units in the LSTM cell is a parameter that can also be tuned in the model to adapt to the particularities of COVID-19 and its variants.

### 3.2. Dataset Description

The model in Figure 1 has been applied to the particular case of the spread of COVID-19 in the Madrid region (Spain). The government of the Madrid region publishes the cumulative incidence data for each health unit (the basic organizational unit which comprises a health center and its influence area) every week. The data are available at [8]. There is a total of 286 health units (each of them with a primary care health center) in the Madrid region.

The incidence data per health unit is divided into 2 different datasets. The first one comprises the early COVID-19 cases from the end of February 2020 until the 1st of July of 2020. The second one covers the cases from the 2nd of July 2020 until the 29th of March 2022. The first days handling the COVID-19 pandemic were characterized by the limited access to PCR tests, and the initial protocols to measure positive cases had to be adjusted over the first months of the pandemic. We have therefore selected the dataset from the 2nd of July 2020 since the procedures used to measure the positive cases have been stable since then until the 29th of March 2022 in which the mechanisms to sense and follow up new cases have been modified following the new global scenario for the sustainable control of COVID-19. The selected period shows the incidence from the 2nd to the 6th wave of the COVID-19 spread.

Figure 3 shows the weekly cases for the region centered at latitude: 40.337718 and longitude: −3.895560. Figure 4 captures the cumulative cases for the same region.

Figure 3 and Figure 4 show the different waves for the COVID-19 spread over the Madrid area, being the last one dominated by the Omicron variant that has shown a significantly higher capacity of infection.

## 4. Results

### 4.1. Parameter Setting for the COVID-19 Forecasting Model

The model in Figure 1 has five major parameters:(1)The number of pixels or areas the Madrid region incidence map is divided into.(2)The size of the input image to the CNN (which is a set of adjacent heath units from the entire image created in the previous point).(3)The number of filters in the CNN.(4)The number of output neurons in the CNN.(5)The number of memory units in the LSTM cell.

In order to estimate the optimal values for the five parameters, the information in [8] has been used to train and validate the model with a range of input values for each parameter. The weekly sensed and reported new cases have been used in order to generate the incidence maps. The relative errors when predicting new COVID-19 weekly cases are higher than the relative errors for cumulative cases and therefore provide a better way to discriminate among different configurations of the model. A 10-fold cross-validation approach has been used. The metric used for validation has been the explained variance (EV), which has been widely used in previous studies such as [19]. The explained variance captures how much of the variance in the forthcoming incidence values are predicted by the model as compared with the variance of the residuals. The EV is a relative measure that is dependent both on the total amount of COVID-19 positive cases and the predicted error. The EV accounts for the proportion of the variance in the predicted variable that can be explained by the model as compared to the variance of the underlying variable. In order to provide normalized figures for EV values, the weekly reported new COVID-19 cases curves have been normalized to have a maximum value of 1 and therefore to be able to compare results among regions with different incidence numbers.

Figure 5 shows the mean values for the EV when training the model with different sizes for the pixel of the generated incidence map image. There are 286 health centers in the Madrid region that have been sensing COVID-19-positive cases during the pandemic. Each of these health centers represents a health zone/unit. If we pixelate a square area encompassing the Madrid region into an *n* by *n* image, the approximate number of pixels that will fall into the Madrid region is 0.55**n^2^*. We have selected four different values for n (16, 20, 24 and 28), which provide average values for health centers per pixel as (2, 1.3, 0.90 and 0.66). The results in Figure 5 show that the best EV score is achieved when using a 16 × 16 input image, which means that each pixel in the image adds the values for two measuring centers on average.

Figure 6 captures the results when using different sizes for the image that is fed into the CNN part of the model. Three different configurations have been tested with images 4 × 4, 6 × 6 and 8 × 8. These images capture the interactions among nearby zones in the incidence map and will be dependent on the local movement of people in the region. In the case of the Madrid region, the optimal EV mean values are achieved for input 6 × 6 images.

The CNN part of the model will try to learn the spatial patterns in the incidence maps by using filters that will be applied using the convolution operation over the input images. The number of filters required will depend on the complexity of the spatial relationships among the pixels in the input image. Figure 7 shows the average results for three different configurations for the number of internal filters (4, 16 and 32). The best values are achieved for the case of 32 filters.

The spatial influence of each image is summarized as the output of the CNN into a set of output neurons. These neurons will be the input for each timestep that will feed the RNN final part of the model in Figure 1. Figure 8 captures the average results for the EV scores for three different configurations for the number of output neurons (2, 8 and 16). The best results are achieved when the number of neurons is 16.

The last parameter in the model is the number of memory units for each LSTM cell in the model in Figure 1. The results for three different configurations (4, 16 and 32) are captured in Figure 9. In this case, the best scores are achieved for 32 memory units.

### 4.2. Validating the COVID-19 Forecasting Model

The model in Figure 1 has been implemented in Python using the Keras library (using TensorFlow) [25]. The graphical representation of the implemented model using the best parameters as described in the previous section is captured in Figure 10. A total of five weekly images are used to capture the pandemic evolution over the last month in order to forecast predictions for the upcoming week. Each input image is first analyzed by 323 × 3 convolutional filters in order to extract the spatial relationships among the pixels in the input image, and then, the information is summarized in two steps. A first summarization step uses a 2 × 2 max pooling layer, and the output is flattened and introduced to a dense fully connected layer. This layer consists of 16 output neurons, which will summarize the temporal contribution of each weekly image to the final RNN layer. The RNN uses LSTM cells with 32 memory units and a final dense layer for predicting the values a week ahead.

Training the model in Figure 10, and using a 10-fold cross validation approach, the new and the cumulative cases one week ahead have been estimated. Figure 11 captures the normalized results for two areas for the new expected cases, and Figure 12 captures the same prediction for the cumulative cases. The relative errors when predicting the cumulative cases are small for large values for cumulative measured cases. We have therefore used the EV for new weekly cases in the previous section.

In order to assess the predictions of the model for a complete wave (not using any data from that wave when training the model), the 10-fold cross-validation has been expanded so that apart from the 10% random samples removed from the training data, an extra set of input data containing all the information in a particular wave has also been removed from the training set. Figure 13 shows the results when removing the data for the 3rd wave when training the model for a particular area in the Madrid region. The results show that the model provides better predictions when the cases are falling back to normality but is not as good in anticipating the appearance of the new wave.

### 4.3. Comparing the COVID-19 Forecasting Model Results with Previous Models

Several previous research studies have published the results for different machine learning models applied to forecasting new COVID-19 cases in different parts of the world such as [19,20]. This section compares the results for two major metrics, the root mean square error (RMSE) and the explained variance (EV), for different models as compared with the model in Figure 1. The summary of results is captured in Table 1.

Table 1 captures the scores for different models and different parts of the world as reported in [19] combined with the scores for our implementation for the best models in [19] applied to the data in [8] and the scores for the model proposed in this manuscript also applied to the data in [8]. The first two lines in Table 1 capture the results for two shallow models (Support Vector Machines and Linear Regression) for predicting the new cases one day ahead in France (data obtained from [19]). The results in [19] improve when using deep learning models such as CNN and LSTM, and the optimal scenario for France is achieved when defining a machine learning model that combines an LSTM and CNN together using the time series for previous cases as the input to the model. Table 1 also shows the results in [19] for the combined LSTM and CNN model to the Indian and the US cases. The results show that bigger populations with a bigger number of cases report higher values for the RMSE score and smaller values for the EV score, which makes the results dependent on the population size. In order to compare the results of the model proposed in Figure 1 with the models captured in [19] as presented in Table 1, we have selected the models showing the best prediction results in [19], implemented them and validated the results for the dataset in [8] in order to provide values for the different models using the same dataset and provide a fair comparison with the performance of the model proposed in this manuscript. The implementation of the combined LSTM and CNN for the time series model in [19] in Python using Keras is shown in Figure 14 (left). The implementation of the LSTM model in [19] in Python using Keras is shown in Figure 14 (right). The internal parameters of the models have been chosen as those reported in [19]. The implemented models have been applied to predict the cases using the dataset in Madrid in order to perform a comparison with the model in Figure 1 using the same dataset. The combined LSTM and CNN for time-series forecasting also outperforms the LSTM model for the data in [8] and achieves an RMSE score of 4.61 and an EV of 0.997. The model in Figure 1 is able to outperform both the RMSE and the EV scores as shown in Table 1, achieving an RMSE value of 1.93 and an EV score of 0.998.

Figure 15 shows the LSTM-CNN implemented model in Figure 14 (left) when applied to forecast the data one week in advance for a particular health area in the dataset.

## 5. Conclusions

The spread of the COVID-19 virus has both a space and time dependency. This paper has proposed a new machine learning model that learns both spatiotemporal patterns based on a sequence of COVID-19 incidence maps. The parameters of the model have been optimized using a dataset for the geolocated 286 health centers in the Madrid (Spain) area and used to forecast the evolution of the pandemic from the second to the sixth waves. The major contribution of the proposed model is the interleaving of the spatial pattern extraction and the time sequence analysis, which work together in optimizing the estimation power of the model.

The results show improved scores in terms of both root mean square error (RMSE) and explained variance (EV) when compared with previous models that have mainly focused on the temporal patterns and dependencies.

The model has been validated both using a 10-fold cross-validation and a combined 10-fold cross-validation with the additional removal of an entire wave of data from the training set in order to assess the results to predict the evolution of new waves. The 10-fold cross-validation provides better results since data from the entire time span is used. When using the model for the prediction of a new wave, the results showed a good fit to the real data for the entire wave except for the beginning of the wave in which the model showed some delay in anticipating the increase in new cases.

As future work, the model will be expanded to use both COVID-19 incidence maps and traffic data in order to use the mobility data to enhance the information available for the model to estimate the spread of the virus.

## Figures and Tables

**Figure 1 sensors-22-03519-f001:**
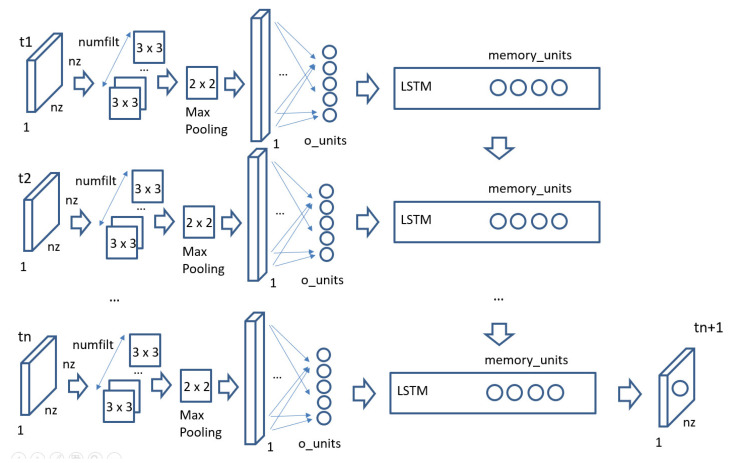
Spatiotemporal CNN-RNN model for COVID-19 forecasting (“…” means the presence of more similar blocks).

**Figure 2 sensors-22-03519-f002:**
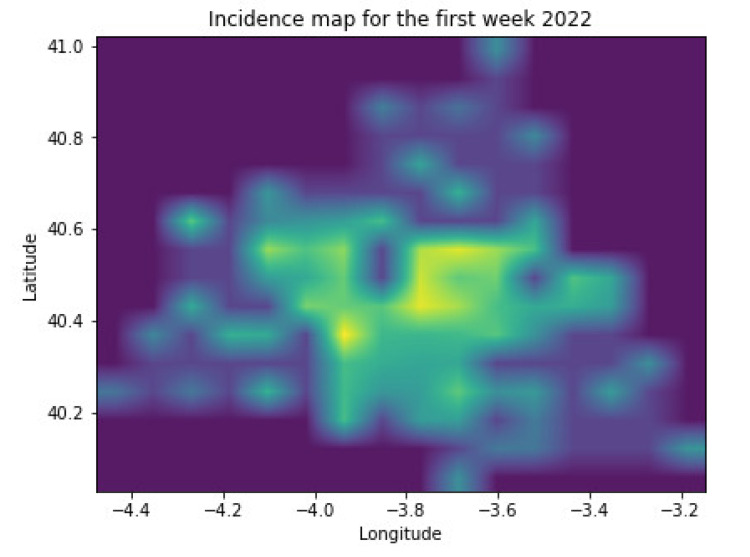
COVID-19 incidence map example in the Madrid region (Spain).

**Figure 3 sensors-22-03519-f003:**
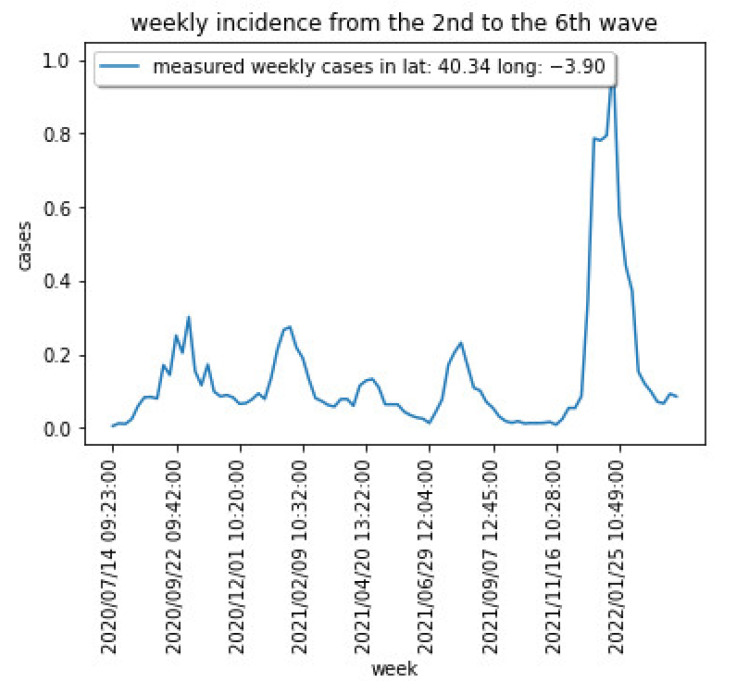
COVID-19 weekly cases for a particular area in the Madrid region.

**Figure 4 sensors-22-03519-f004:**
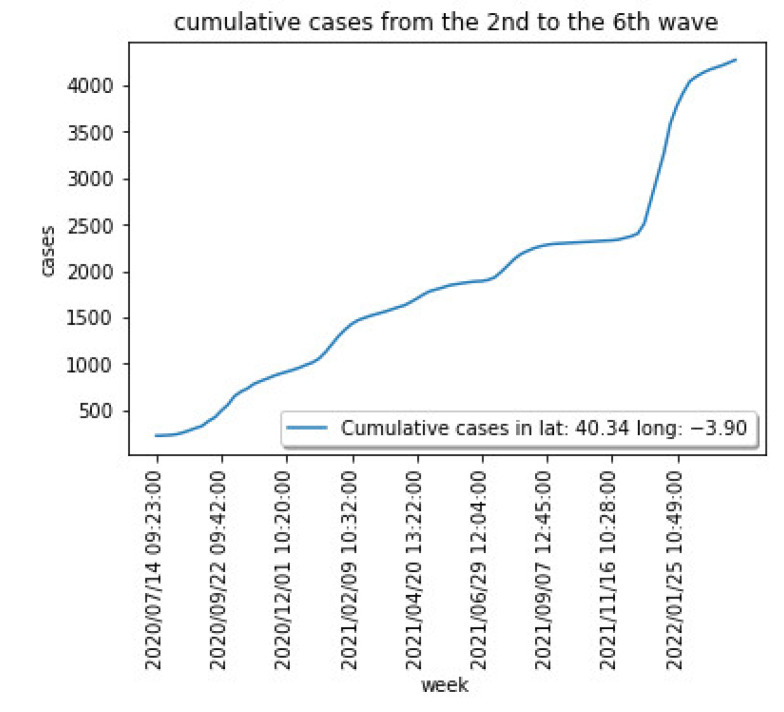
COVID-19 cumulative cases for a particular area in the Madrid region.

**Figure 5 sensors-22-03519-f005:**
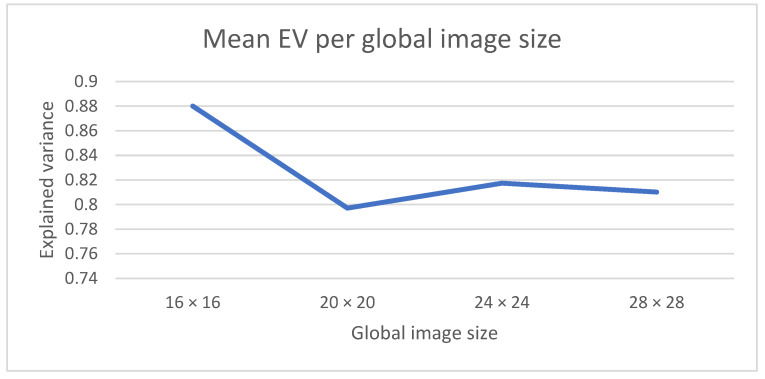
Number of pixels into which the Madrid region map is divided.

**Figure 6 sensors-22-03519-f006:**
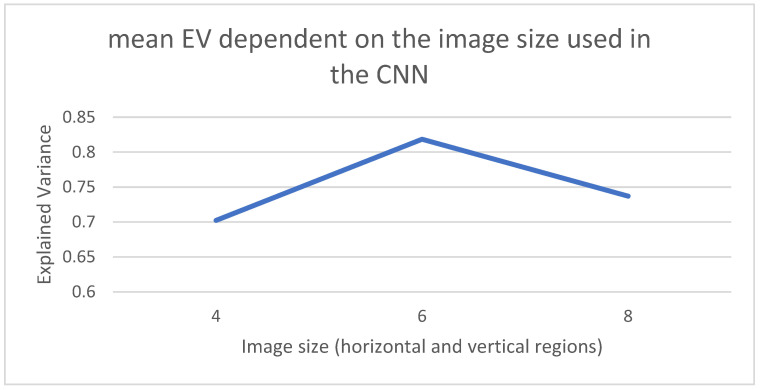
Size of the input image to the CNN.

**Figure 7 sensors-22-03519-f007:**
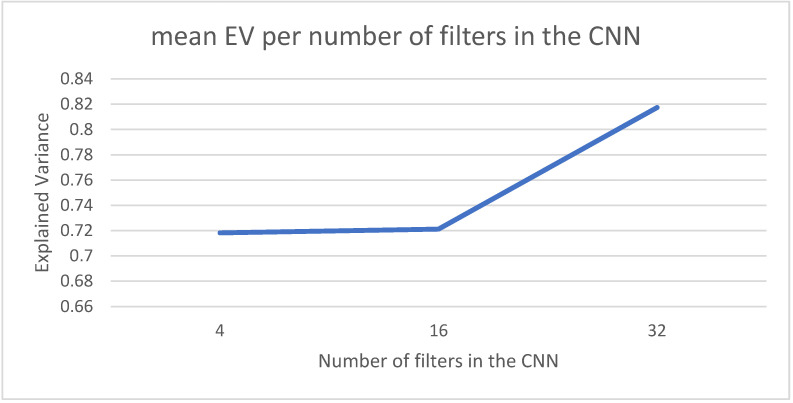
Number of filters used in the first layer by the CNN part of the model.

**Figure 8 sensors-22-03519-f008:**
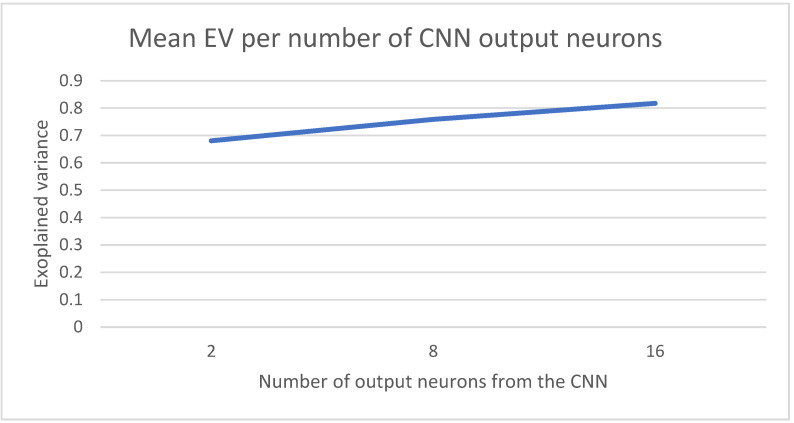
Number of output neurons into which the information processed by the CNN is divided.

**Figure 9 sensors-22-03519-f009:**
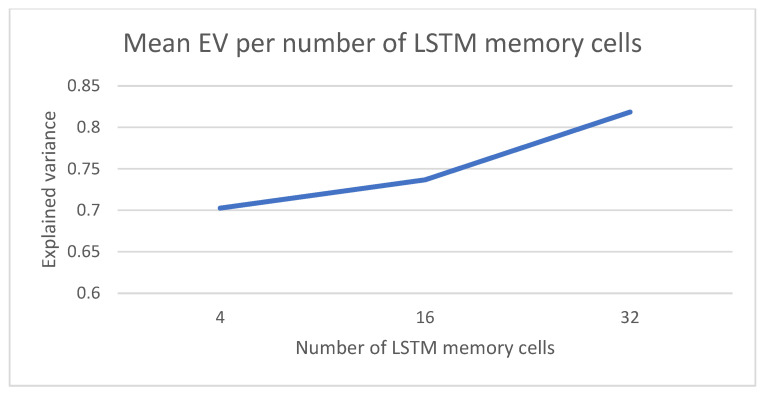
Number of memory units in each LSTM cell for the RNN part of the model.

**Figure 10 sensors-22-03519-f010:**
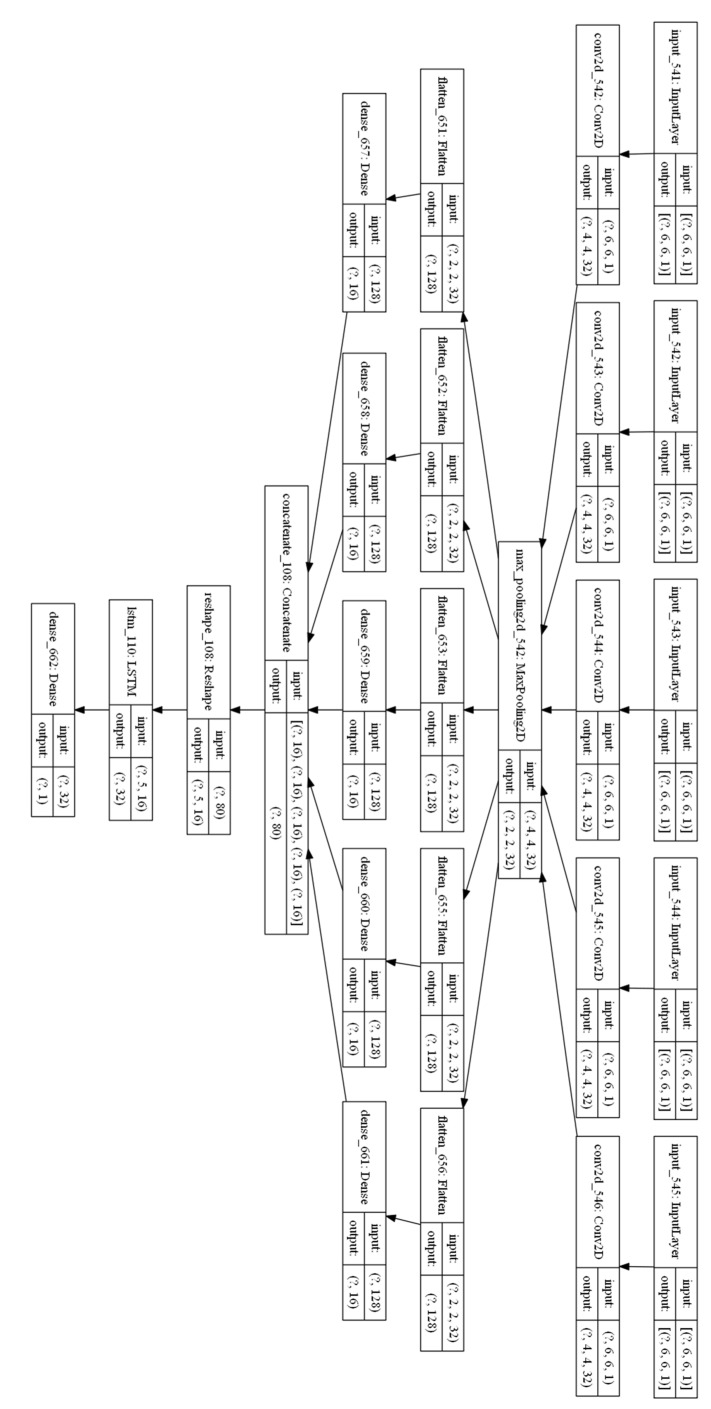
Model graphical representation showing the selected values for the model parameters.

**Figure 11 sensors-22-03519-f011:**
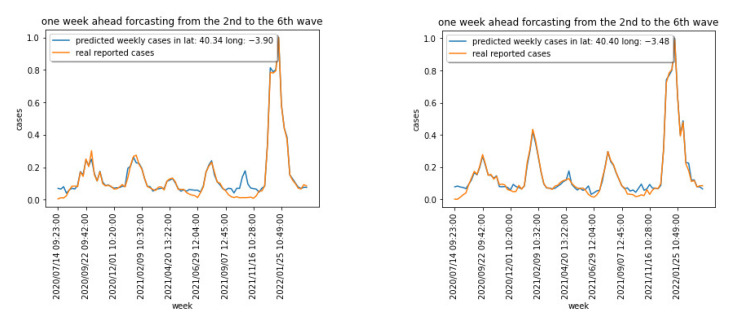
Normalized predicted new cases for 2 areas.

**Figure 12 sensors-22-03519-f012:**
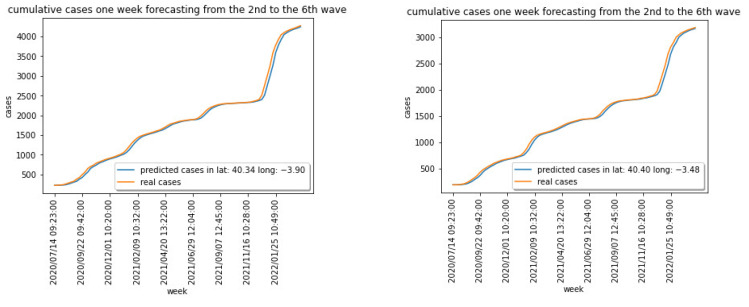
Normalized predicted cumulative cases for 2 areas.

**Figure 13 sensors-22-03519-f013:**
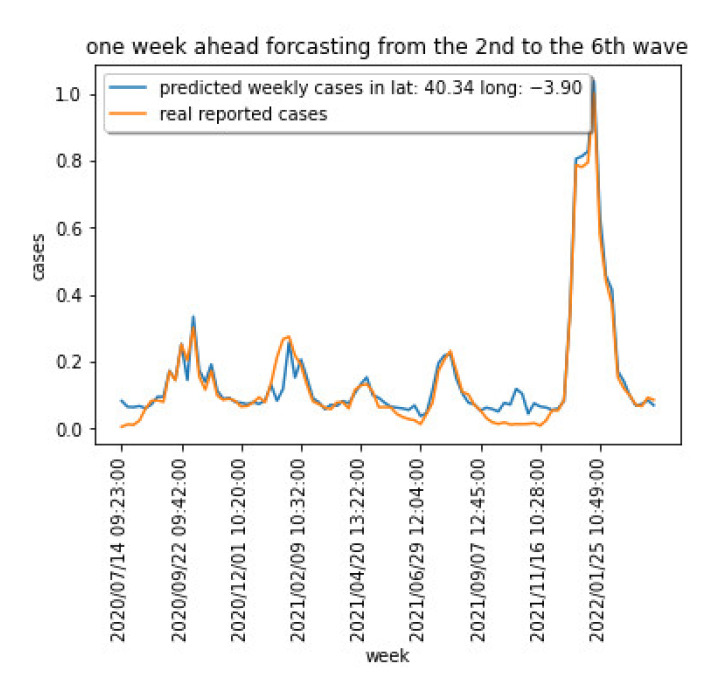
Predicting the values for the 3rd wave.

**Figure 14 sensors-22-03519-f014:**
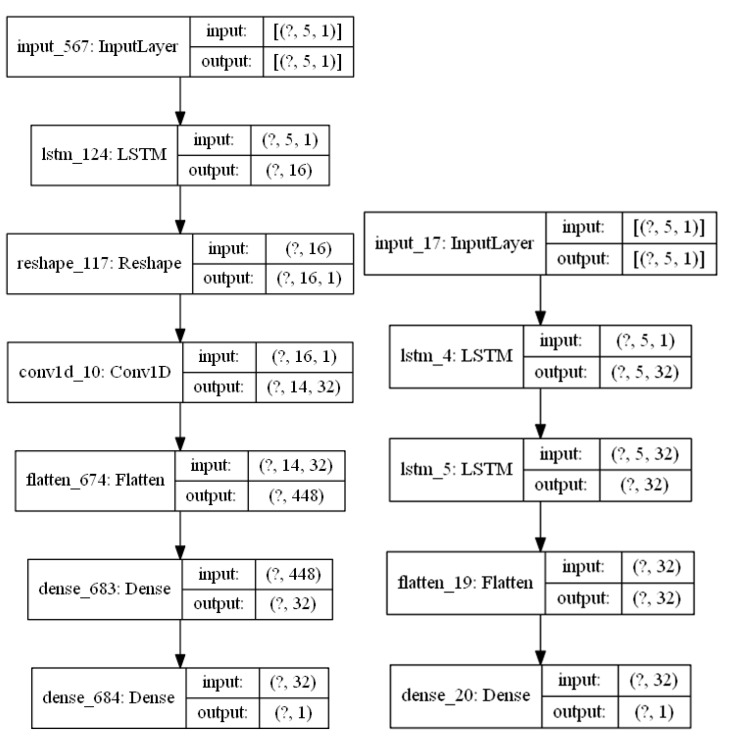
Our implementation of the LSTM-CNN model in [19] (**left**) and of the LSTM model in [19] (**right**).

**Figure 15 sensors-22-03519-f015:**
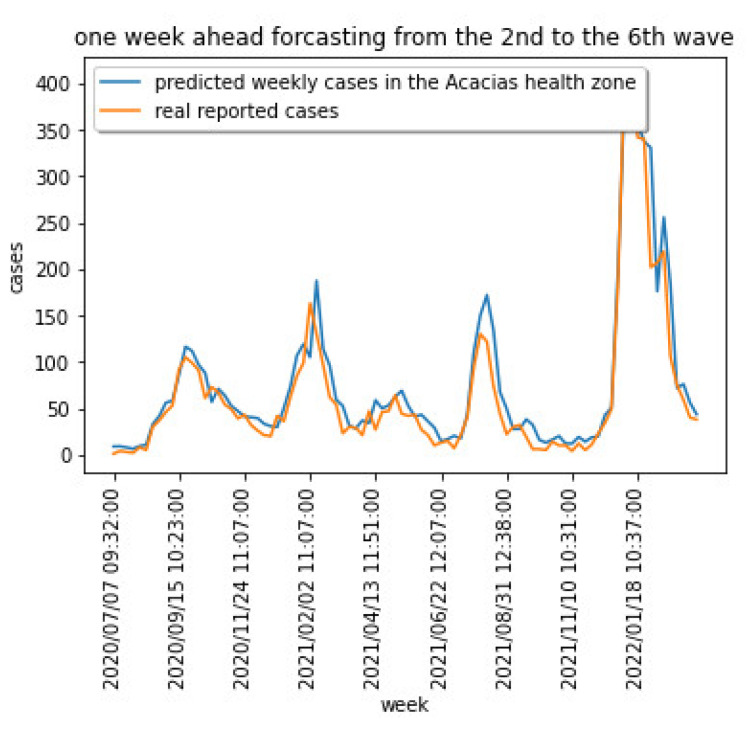
Predicting one week ahead new cases using the time-series LSTM-CNN model in [19] for the Acacias health zone data.

**Table 1 sensors-22-03519-t001:** RMSE and EV values for different machine learning models applied to data in different parts of the world.

Model	RMSE	EV
SVM ^1^ (France) [19]	6560 × 10^6^	0.892
LR ^2^ (France) [19]	5210 × 10^6^	0.810
RBM (France) [19]	8540	0.957
CNN (France) [19]	1930 × 10^6^	0.975
LSTM (France) [19]	2180 ×10^6^	0.967
LSTM (Madrid)	19.71	0.996
Time series LSTM-CNN (France) [19]	2750	0.994
Time series LSTM-CNN (India) [19]	83,100	0.998
Time series LSTM-CNN (US) [19]	56,900	0.999
Time series LSTM-CNN (Madrid)	4.61	0.997
**Spatiotemporal CNN+LSTM (Madrid areas)**	**1.93**	**0.998**

^1^ Support Vector Machine. ^2^ Linear Regression. In bold the results for our proposed model.

## Data Availability

Data supporting reported results can be found at Comunidad de Madrid. COVID-19 Open data by basic health care zones. Available at: https://datos.comunidad.madrid/catalogo/dataset/covid19_tia_zonas_basicas_salud. Last accessed on 2 May 2022.

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
