# Peer review of "Deep Spatiotemporal Model for COVID-19 Forecasting"

_sensors, 2022, doi:10.3390/s22093519_

Round 1

Reviewer 1 Report

The study aims to tackle the spread of COVID-19 by combining the LSTM and RNN models to model the temporal and spatial pattern of its change. But it still has some problems in terms of the structure and description. I recommend a major revision. My comments are as follows.

In the introduction section, the authors need to state the gap of existing studies and the contributions of the manuscript clearly. For example, in the second paragraph, the authors introduced different types of models to forecast COVID-19 infections. However, there are no summaries about these methods. The last sentence seems to tell the reader that the hybrid models are the best selection to forecast the infected cases. However, it seems that the manuscript does not take the hybrid method.

In the related work section, besides the application of machine learning models, the authors also need to introduce the studies related to the impact of COVID-19 on society and applications of deep learning methods in other areas.

The following references maybe help.

  • https://www.sciencedirect.com/science/article/pii/S0198971521001101
  • https://www.sciencedirect.com/science/article/pii/S2772424721000123?via%3Dihub
  • https://www.emerald.com/insight/content/doi/10.1108/JICV-03-2021-0004/full/html

Figure 2 seems like a KDE map. But according to the description, the data is the aggregation of infected cases at a grid level. A more detailed introduction of the data should be provided.

The title of Table 1 should be updated. In addition, it seems the comparison of different models is applied in multi-areas. However, these data are not introduced in the previous section. Furthermore, the compared models are from reference [19], did you implement them by yourself?

The authors need to refine the language of the paper. For example, SARS-COV-.2 virus -> SARS-COV-2 virus in the abstract.

Author Response

All the comments from the reviewers have been thoroughly and carefully implemented in order to improve the manuscript. We want to thank the reviewers for their constructive comments.

Reviewer 1:

The study aims to tackle the spread of COVID-19 by combining the LSTM and RNN models to model the temporal and spatial pattern of its change. But it still has some problems in terms of the structure and description. I recommend a major revision. My comments are as follows.

Comment: 

In the introduction section, the authors need to state the gap of existing studies and the contributions of the manuscript clearly. For example, in the second paragraph, the authors introduced different types of models to forecast COVID-19 infections. However, there are no summaries about these methods. The last sentence seems to tell the reader that the hybrid models are the best selection to forecast the infected cases. However, it seems that the manuscript does not take the hybrid method.

Response:

We want to thank the reviewer for the comment. We have provided summaries about the methods cited in the second paragraph of the Introduction section and highlighted some limitations in the hybrid models in order to stat the gap of existing studies and the contributions of the manuscript. The paragraph has been modified as follows:

Different types of predictive models have been used in order to forecast COVID-19 infections, recoveries and deaths such as epidemic, mathematical, statistical, machine learning based and hybrid models. The most applied epidemic models are the Susceptible, Infected, and Recovered (SIR) model and the Susceptible, Exposed, Infected, and Recovered (SEIR) model [4]. Both SIR and SEIR models group people in different compartments and solve differential equations to move people among compartments, modelling the spread of the disease. These differential equations describe the variation over time of the amount of people that get infected after being exposed to the virus and finally either recover or die from it based on the amount of people susceptible and already infected and the dynamics of the virus captured in rate parameters. Statistical models describe the spread of the disease in terms of stochastic variables which can be modeled using probability functions, some of which can be observed while others can be estimated. The authors in [5] describe the concept of statistical models for COVID-19 and use data to fit probability distributions to stochastic variables defining the spread of the COVID-19 virus such as the time to develop symptoms and the time to require hospitalization. Machine learning models try to learn patterns in the observed data by training models that can learn from data. Once trained, the models could be used to predict the outcome for new data not currently seen by the model. The authors in [6] published a survey on several machine learning models that have been used for COVID-19 predictions including deep learning models. Compared with epidemic models, machine learning models do not have to generate a simplified system in order to characterize the spreading of the virus but to observe data samples coming from different sensors and learn from them. However, machine learning models require to have previous data in order to fit the weights inside them. This is even more significant for deep machine learning models which are able to learn more intricate patterns from data samples but have to find optimal values for a bigger number of weights which require to have enough data to train them. Epidemic and machine learning models have also been combined into hybrid models which use machine learning models in order to find values from data for the different parameters in the epidemic models. The authors in [7] implemented and evaluated a framework with machine learning models trained to extract epidemic dynamics from the infection data to improve a county-level spatiotemporal epidemiological model that combines a spatial Cellular Automaton (CA) with a temporal Susceptible-Undiagnosed-Infected-Removed (SUIR) model. Although hybrid models are able to tune and optimize results for epidemic models they will require the availability of data and will be limited to the simplifications made by the epidemic model.       

Comment: 

In the related work section, besides the application of machine learning models, the authors also need to introduce the studies related to the impact of COVID-19 on society and applications of deep learning methods in other areas.

The following references maybe help.

  • https://www.sciencedirect.com/science/article/pii/S0198971521001101
  • https://www.sciencedirect.com/science/article/pii/S2772424721000123?via%3Dihub
  • https://www.emerald.com/insight/content/doi/10.1108/JICV-03-2021-0004/full/html

Response:

Very positive comment. In fact, we have plans to add traffic estimation models to enhance the COVID-19 predictions in future work and adding these references has allowed us both to introduce the studies related to the impact of COVID-19 on society and to motivate future works.

We have added the 3 references:

  1. Li, A., Zhao, P., Haitao, H., Mansourian, A., & Axhausen, K. W. (2021). How did micro-mobility change in response to COVID-19 pandemic? A case study based on spatial-temporal-semantic analytics. Computers, Environment and Urban Systems, 90, 101703.
  2. Liu, Y., Lyu, C., Zhang, Y., Liu, Z., Yu, W., & Qu, X. (2021). DeepTSP: Deep traffic state prediction model based on large-scale empirical data. Communications in Transportation Research, 1, 100012.
  3. Zhu, W., Wu, J., Fu, T., Wang, J., Zhang, J., & Shangguan, Q. (2021). Dynamic prediction of traffic incident duration on urban expressways: a deep learning approach based on LSTM and MLP. Journal of intelligent and connected vehicles.

We have also added the following paragraph to section 2:

The spread of the COVID-19 virus has affected different behavioral aspects of society. Lockdowns and social policies have affected areas such as mobility in order to minimize social interactions. The authors in [21] analyzed the changes in micro-mobility usage before and during the lockdown period exploiting high-resolution micro-mobility trip data collected in Zurich, Switzerland. Specifically, docked bike, docked e-bike, and dockless e-bike were evaluated and compared from the perspective of space, time and semantics. The authors showed that the spread of the COVID-19 virus affected mobility policies and at the same time, modified mobility patterns affected the spread of the virus. There are some studies such as [22] and [23] that use machine learning models in order to estimate traffic patterns which in turn could be used to enhance COVID-19 predictions. The study in [22] focuses on the construction of an effective solution designed for spatio-temporal data to predict the traffic state of large-scale traffic systems while the authors in [23] proposed a framework based on the multi-layer perception (MLP) and long short-term memory (LSTM) model which integrate traffic incident-related factors and real-time traffic flow parameters to predict the residual traffic incident duration. Adding mobility data to the model proposed in this manuscript will be described in future publications.    

Comment: 

Figure 2 seems like a KDE map. But according to the description, the data is the aggregation of infected cases at a grid level. A more detailed introduction of the data should be provided.

Response:

The following text has been added to provide more details of the data:

The data in [8] provides information about the new cases reported weekly by each primary care health center. There are 286 primary care health centers in the Madrid region and each of them covers a particular area in the region. Figure 2 shows the mapping of the 286 health areas into a grid image

Comment: 

The title of Table 1 should be updated. In addition, it seems the comparison of different models is applied in multi-areas. However, these data are not introduced in the previous section. Furthermore, the compared models are from reference [19], did you implement them by yourself?

Response:

We have updated the title of the table:

Table 1. RMSE and EV values for different machine learning models applied to data in different parts of the world

We have used the results in [19] which are applied to different areas of the World as described in the reference but not applied to our dataset in Madrid as described in [8]. Moreover, we have selected the best models in [19] and implemented ourselves and applied them to the dataset in [8]. We have added the results to table 1 and Figures 14 and 15.

The following text has therefore been updated:

Table 1 shows the scores for different models and different parts of the world as reported in [19]. The first two lines in Table 1 capture the results for two shallow models (Support Vector Machines and Linear Regression) for predicting the new cases one day ahead in France (data obtained from [19]). The results in [19] improve when using Deep Learning Models such as CNN and LSTM and the optimal scenario for France is achieved when defining a machine learning models that combines and LSTM and CNN together using the time series for previous cases as the input to the model. Table 1 also shows the results in [19] for the combined LSTM and CNN model to the Indian and the US cases. The results show that bigger populations with a bigger number of cases report higher values for the RMSE score and smaller values for the EV score which makes the results dependent on the population size. In order to compare the results of the model proposed in Figure 1 with the models captured in [19] as presented in Table 1, we have selected the models showing best prediction results in [19], implemented them and validate the results for the dataset in [8] in order to provide values for the different models using the same dataset and provide a fair comparison with the performance of the model proposed in this manuscript. The implementation of the combined LSTM and CNN for time series model in [19] in Python using Keras is shown in Figure 14 (left). The implementation of the LSTM model in [19] in Python using Keras is shown in Figure 14 (right). The internal parameters of the models have been chosen as those reported in [19]. The implemented models have been applied to predict the cases using the dataset in Madrid in order to perform a comparison with the model in Figure 1 using the same dataset. The combined LSTM and CNN for time series forecasting also outperforms the LSTM model for the data in [8] and achieves a RMSE score of 4.61 and an EV of 0.997. The model in Figure 1 is able to outperform both the RMSE and the EV scores as shown in Table 1, achieving a RMSE value of 1.93 and an EV score of 0.998.

Table 1. RMSE and EV values for different machine learning models applied to data in different parts of the world.

Model

RMSE

EV

SVM1 (France) [19]

6560*106

0.892

LR2 (France) [19]

5210*106

0.810

RBM (France) [19]

8540

0.957

CNN (France) [19]

1930*106

0.975

LSTM (France) [19]

2180*106

0.967

LSTM (Madrid)

19.71

0.996

Time series LSTM-CNN (France) [19]

2750

0.994

Time series LSTM-CNN (India) [19]

83100

0.998

Time series LSTM-CNN (US) [19]

56900

0.999

Time series LSTM-CNN (Madrid)

4.61

0.997

Spatiotemporal CNN+LSTM (Madrid areas)

1.93

0.998

1 Support Vector Machine. 2Linear Regression

              << Image can not be pasted here but shown in the attached file>>

Figure 14. LSTM-CNN model in [19] implementation (left) and LSTM model in [19] implementation (right)

Comment: 

The authors need to refine the language of the paper. For example, SARS-COV-.2 virus -> SARS-COV-2 virus in the abstract.

Response:

Done.

Reviewer 2 Report

The model generates sequences of infection maps from georeferenced COVID-19 reported data and makes use of both CNN and RNN to estimate the spread of the virus for each region in the map. The model is validated using the sensed and reported data from the Madrid Region in Spain.
Above some recommendations to enhance the quality of the manuscript:

1- Title of Table 1 is not correct.

2- The proposed CNN-RNN approach should be compared with the methods cited in Table1 using the same data in [8].
This is important to evaluate the performance and behavior of the suggested model compared to other algorithms.

3- Please determine accurately the contribution of the paper, compared to the state of the art, regarding the design of the used CNN-RNN model. 

4- In the same context of yours, you can refer in the state of the art to this study proposing a simulation model for Forecasting COVID-19: https://doi.org/10.3390/su13094888

5- The conclusion: Please mention the used approach and contributions before illustrating the findings of the results.

6- The conclusion: Please add future directions/perspectives of the study.

Author Response

All the comments from the reviewers have been thoroughly and carefully implemented in order to improve the manuscript. We want to thank the reviewers for their constructive comments.

Reviewer 2:

The model generates sequences of infection maps from georeferenced COVID-19 reported data and makes use of both CNN and RNN to estimate the spread of the virus for each region in the map. The model is validated using the sensed and reported data from the Madrid Region in Spain.
Above some recommendations to enhance the quality of the manuscript:

Comment: 

1- Title of Table 1 is not correct.

Response:

We have updated the title of the table:

Table 1. RMSE and EV values for different machine learning models applied to data in different parts of the world

Comment: 

2- The proposed CNN-RNN approach should be compared with the methods cited in Table1 using the same data in [8].
This is important to evaluate the performance and behavior of the suggested model compared to other algorithms.

Response:

Thank you very much for the recommendation. We have selected the best models in [19] and implemented ourselves and applied them to the dataset in [8]. We have added the results to table 1 and Figures 14 and 15.

The following text has therefore been updated:

Table 1 shows the scores for different models and different parts of the world as reported in [19]. The first two lines in Table 1 capture the results for two shallow models (Support Vector Machines and Linear Regression) for predicting the new cases one day ahead in France (data obtained from [19]). The results in [19] improve when using Deep Learning Models such as CNN and LSTM and the optimal scenario for France is achieved when defining a machine learning models that combines and LSTM and CNN together using the time series for previous cases as the input to the model. Table 1 also shows the results in [19] for the combined LSTM and CNN model to the Indian and the US cases. The results show that bigger populations with a bigger number of cases report higher values for the RMSE score and smaller values for the EV score which makes the results dependent on the population size. In order to compare the results of the model proposed in Figure 1 with the models captured in [19] as presented in Table 1, we have selected the models showing best prediction results in [19], implemented them and validate the results for the dataset in [8] in order to provide values for the different models using the same dataset and provide a fair comparison with the performance of the model proposed in this manuscript. The implementation of the combined LSTM and CNN for time series model in [19] in Python using Keras is shown in Figure 14 (left). The implementation of the LSTM model in [19] in Python using Keras is shown in Figure 14 (right). The internal parameters of the models have been chosen as those reported in [19]. The implemented models have been applied to predict the cases using the dataset in Madrid in order to perform a comparison with the model in Figure 1 using the same dataset. The combined LSTM and CNN for time series forecasting also outperforms the LSTM model for the data in [8] and achieves a RMSE score of 4.61 and an EV of 0.997. The model in Figure 1 is able to outperform both the RMSE and the EV scores as shown in Table 1, achieving a RMSE value of 1.93 and an EV score of 0.998.

Table 1. RMSE and EV values for different machine learning models applied to data in different parts of the world.

Model

RMSE

EV

SVM1 (France) [19]

6560*106

0.892

LR2 (France) [19]

5210*106

0.810

RBM (France) [19]

8540

0.957

CNN (France) [19]

1930*106

0.975

LSTM (France) [19]

2180*106

0.967

LSTM (Madrid)

19.71

0.996

Time series LSTM-CNN (France) [19]

2750

0.994

Time series LSTM-CNN (India) [19]

83100

0.998

Time series LSTM-CNN (US) [19]

56900

0.999

Time series LSTM-CNN (Madrid)

4.61

0.997

Spatiotemporal CNN+LSTM (Madrid areas)

1.93

0.998

1 Support Vector Machine. 2Linear Regression

              <<image can not be pasted here but included in the attached file>>

Figure 14. LSTM-CNN model in [19] implementation (left) and LSTM model in [19] implementation (right)

Comment: 

3- Please determine accurately the contribution of the paper, compared to the state of the art, regarding the design of the used CNN-RNN model. 

Response:

We have added the following text to the introduction:

The major contribution of the proposed model is that is able to use the combined space and time information together over a region outperforming previous models found in literature when using the same dataset.

This statement is fully detailed in section 4 where we have added the following text:

In order to compare the results of the model proposed in Figure 1 with the models captured in [19] as presented in Table 1, we have selected the models showing best prediction results in [19], implemented them and validate the results for the dataset in [8] in order to provide values for the different models using the same dataset and provide a fair comparison with the performance of the model proposed in this manuscript. The implementation of the combined LSTM and CNN for time series model in [19] in Python using Keras is shown in Figure 14 (left). The implementation of the LSTM model in [19] in Python using Keras is shown in Figure 14 (right). The internal parameters of the models have been chosen as those reported in [19]. The implemented models have been applied to predict the cases using the dataset in Madrid in order to perform a comparison with the model in Figure 1 using the same dataset. The combined LSTM and CNN for time series forecasting also outperforms the LSTM model for the data in [8] and achieves a RMSE score of 4.61 and an EV of 0.997. The model in Figure 1 is able to outperform both the RMSE and the EV scores as shown in Table 1, achieving a RMSE value of 1.93 and an EV score of 0.998.

Comment: 

4- In the same context of yours, you can refer in the state of the art to this study proposing a simulation model for Forecasting COVID-19: https://doi.org/10.3390/su13094888

Response:

We have cited the referenced publication which has allowed us to introduce and link the simulator based models with the work in the current paper. The following text has been added to section 2:

As previously captured in the introduction section, a different method for COVID-19 forecasting is based on the use of epidemic models. These models have been implemented in simulators such as [24] which introduced a simulation-based model to forecast the spreading behavior of the COVID-19 based on Saudi real data. The model we propose in this manuscript will also be enhanced to help with the estimation of the underlying parameters required to tune COVID-19 simulators as a future publication.     

And the following reference:

  1. Hassanat, A. B., Mnasri, S., Aseeri, M., Alhazmi, K., Cheikhrouhou, O., Altarawneh, G., ... & Almoamari, H. (2021). A simulation model for forecasting covid-19 pandemic spread: Analytical results based on the current saudi covid-19 data. Sustainability, 13(9), 4888.

Comment: 

5- The conclusion: Please mention the used approach and contributions before illustrating the findings of the results.

Response:

The first paragraph in the conclusion section has been expanded to include this comment as follows:

The spread of the COVID-19 virus has both a space and time dependency. This paper has proposed a new machine learning model that learns both spatiotemporal patterns based on a sequence of COVID-19 incidence maps. The parameters of the model have been optimized using a dataset for the geolocated 286 health centers in the Madrid (Spain) area and used to forecast the evolution of the pandemic from the second to the sixth waves. The major contribution of the proposed model is the interleaving of the spatial pattern extraction and the time sequence analysis which work together in optimizing the estimation power of the model. 

Comment: 

6- The conclusion: Please add future directions/perspectives of the study.

Response:

We have added the following paragraph to the end of the conclusion section:

As a future work, the model will be expanded to use both COVID-19 incidence maps and traffic data in order to use the mobility data to enhance the information available for the model to estimate the spread of the virus.

Reviewer 3 Report

This paper proposes a deep learning model for COVID-19 forecasting that combines spatiotemporal information to estimate how the virus will migrate inside an area over time. It’s an interesting paper that is well presented and structured. Therefore, I would suggest accepting after minor revisions.

  • Abstract: I would expect to read a short sentence on the results/ conclusion.
  • How did you select the region centre?
  • Figure 1: remove the red underlines
  • Line 231: 0.55*n2 à what’s n? How did you select the four values?
  • Are there other RMSE of other spatiotemporal approaches to compare the accuracy of your suggested model?
  • Table 1: revise the caption of the table.
  • Figure 14: revise the caption. If I understand correctly, you show how you implement your model using the parameters from other authors, right? If so, the current Figure caption makes the readers understand that you use other authors’ figure.
  • Why do you think the model didn’t have a good fit at the beginning of the wave? How could this be improved?
  • Could this model be used for shaping policies and/ or strategies against covid? If so, under which circumstances?

Author Response

All the comments from the reviewers have been thoroughly and carefully implemented in order to improve the manuscript. We want to thank the reviewers for their constructive comments.

Reviewer 3:

This paper proposes a deep learning model for COVID-19 forecasting that combines spatiotemporal information to estimate how the virus will migrate inside an area over time. It’s an interesting paper that is well presented and structured. Therefore, I would suggest accepting after minor revisions.

Comment: 

  • Abstract: I would expect to read a short sentence on the results/ conclusion.

Response:

The following sentence has been added to the abstract:

The results show improved scores in terms of both Root Mean Square Error (RMSE) and Explained Variance (EV) when compared with previous models that have mainly focused on the temporal patterns and dependencies.

Comment: 

  • How did you select the region centre?

Response:

The image grid limits were selected to cover the entire Comunidad de Madrid region. Depending on the number of pixels in the generated images, the quadrants were computed using the following Python method:

def quadrant(nc, lat, long, mala, malo, mila, milo):

    cx=int((nc-1)*(long-milo)/(malo-milo))

    cy=int((nc-1)*(lat-mila)/(mala-mila))

    return cx,cy

Comment: 

  • Figure 1: remove the red underlines

Response:

Removed. The figure is now:

<<figure can not be pasted here but included in the attached file>>

Comment: 

  • Line 231: 0.55*n2à what’s n? How did you select the four values?

Response:

The number of pixels in each dimension. The following text is now explaining it:

we pixelate a square area encompassing the Madrid region into an n by n image

Comment: 

  • Are there other RMSE of other spatiotemporal approaches to compare the accuracy of your suggested model?

Response:

There are metrics such as the Root Mean Squared Error (RMSE), the Mean absolute error (MAE), the R-Squared (R2), the explained variance (EV)   and the mean absolute percentage error (MAPE) that have been previously used in literature. We have selected 2 of them which are used in previous papers that are used to compared results with those in the current paper: the Root Mean Square Error (RMSE) and the Explained Variance (EV)

Comment: 

  • Table 1: revise the caption of the table.

Response:

Done

Comment: 

  • Figure 14: revise the caption. If I understand correctly, you show how you implement your model using the parameters from other authors, right? If so, the current Figure caption makes the readers understand that you use other authors’ figure.

Response:

The caption has been changed to:

Figure 14. Our implementation of the LSTM-CNN model in [19] (left) and of the LSTM model in [19] (right)

Comment: 

  • Why do you think the model didn’t have a good fit at the beginning of the wave? How could this be improved?

Response:

This is a common aspect in previous models. When there is a residual number of cases it is difficult to anticipate the start of a new wave from it. Once the cases start to grow for a new wave, the model uses previous data to estimate how fast the cases will scale. In our opinion, this could be enhanced if a longer period of data was used to feed the model so that the periodicity between waves can be learnt. However, a longer period of time would be required to train the model that the one currently available.

Comment: 

  • Could this model be used for shaping policies and/ or strategies against covid? If so, under which circumstances?

Response:

In our opinion, this is one of the applications of the model. Using real data, the model is able to predict how fast the cases will grow in the near future which could help in anticipating the introduction of social policies. Using synthetic data, the model could be used to simulate scenarios for new waves which could also be used to help public authorities in their policy making.

Round 2

Reviewer 1 Report

Thank you for your efforts. I have no more comments.